# Incorporating wellbeing into general factor models: A more complete mental state?

**Ritika Chokhani** [1]*, **Suzet Tanya Lereya**[2], **Jessica Deighton**[1,2]

**1** Division of Psychology and Language Sciences, University College London, London, United Kingdom,
**2** Evidence-Based Practice Unit, University College London and Anna Freud, London, United Kingdom

\* ritika.chokhani.14@ucl.ac.uk

## Abstract

### Purpose

This study aimed to understand whether incorporating wellbeing as another dimension within general factor models of mental health is (a) feasible and (b) useful.

### Methods

Data from two time points (Year 7 and Year 9) for 15258 adolescents who participated in the HeadStart programme in England was used. In Stage 1, we used structural equation modelling on time point 1 data to test different latent variable models incorporating psychopathology and wellbeing dimensions. In Stage 2, we tested whether the latent factors identified in Stage 1 significantly predicted impairment at time point 2.

### Results

A general factor model incorporating a shared underlying dimension between (lack of) wellbeing and psychopathology as well as unique specific factors had good fit to the data at Stage 1. Further, although both general factor models with and without wellbeing fit the data well at Stage 1, only the general factor model with wellbeing met all required fit thresholds when regressions to predict impairment were added in. The model without any general factor (correlated factors model) met pre-defined fit thresholds but had lower fit indices.

### Conclusion

The incorporation of wellbeing into general factor models may help represent more nuanced mental health states and may be useful in predicting future functional states, however such a model needs further replication with comprehensive measures and comparison with alternative models to verify its validity and utility.

**Data availability statement:** An anonymised and archived version of the dataset without 4 sociodemographic variables is available through the UK Data Service at: http://doi.org/10.5255/UKDA-SN-9150-1 The code for this study is available at: https://osf.io/2egv9/.

**Funding:** R.C. is supported by the Wellcome Trust (218497/Z/19/Z). Wellcome Trust played no role in the study design, data collection and analysis, decision to publish and preparation of this manuscript. The data in this study was collected as part of the HeadStart learning programme and supported by funding from The National Lottery Community Fund. The content is solely the responsibility of the authors, and it does not reflect the views of The National Lottery Community Fund.

**Competing interests:** The authors have declared that no competing interests exist.

## Introduction

General factor modelling of mental health has gained traction in the last two decades, perhaps best exemplified in research on a general factor of psychopathology or the 'p factor' [1,2]. For proponents, such general factor modelling appears to have advantages on both theoretical and empirical fronts. Theoretically, it represents a transdiagnostic rather than a disorder-specific model of mental ill health, which better explains high rates of comorbidity in mental disorder diagnoses and better accounts for individuals waxing and waning through symptoms over a time period [3]. Empirically, it appears to strongly predict outcomes as diverse as future psychopathology [4], academic attainment and school functioning [4,5], criminal convictions [6] and suicide attempts [1]. The p factor is also related to common constructs in the nomological networks of mental health disorders, including socioemotional and cognitive constructs [7–10]. Although the p factor's substantive meaning is strongly contested (see [11] and [12] for a review), its consistent identification across different populations, samples, measures and statistical models [4–6,8–10,13–16], its stability and homotypic continuity [17,18] and its empirical association with variables of interest remains formidable [8–10].

Even if general factors do not represent etiologically substantive constructs, they model mental health status in a transdiagnostic and dimensional manner, as compared to the categorical diagnoses approach [19]. In this study, we conceptualize general factors as useful indices of the general mental health status of an individual on a continuum. As such, they may have utility in predicting future mental or functional states or other related outcomes. Further, from a measurement perspective, general factor models incorporating specific factors are a useful way to understand and model the different constructs that should be considered when measuring mental health status and to test the relationship of these constructs to each other.

Bifactor models have commonly been used to model general factors of mental health, often including a general factor dimension as well as specific internalizing and externalizing dimensions, with additional specific psychopathology dimensions representing thought disorder, hyperactivity and substance use variably added [1,14,18]. We argue that wellbeing could be another dimension in such a general factor model. In 2005, Keyes proposed the complete state model, which argued that mental illness and mental health are distinct, but related constructs [20]. This model conceptualized the presence of mental health as the presence of emotional, psychological and subjective wellbeing which may co-occur with both the presence and absence of mental disorder [20,21]. This model has been empirically tested using three different approaches. First, Confirmatory Factor Analysis (CFA) models have found an oblique two-factor model having the best fit to data, suggesting that wellbeing and psychopathology are distinct but correlated constructs (see [22] for a review). Second, many studies have found unique as well as shared predictors of wellbeing and mental illness [23–26], lending further support to the idea of distinct constructs with some overlap. Third, studies have classified individuals into quadrants based on combinations of high/low mental illness and high/low wellbeing; two out of these subgroups

would typically not be represented in unidimensional psychopathology models (those with low wellbeing but no mental illness; those with mental illness but reporting high wellbeing). These studies have indeed found distinct relationships between being situated in a particular quadrant with both outcome and predictor variables. Notably, having low wellbeing even in the absence of mental illness predisposes one to worse functioning and poorer resilience [20,27]. Similarly, complete mental health is associated with better functioning than just low psychopathology [28]. While both the quadrant approach and the dual-factor CFA approach appropriately tend to highlight the distinctness of wellbeing and psychopathology, they may downplay their shared aspects and may not fully represent the complex relationship between wellbeing and psychopathology [29].

Including wellbeing in general factor models along with psychopathology could address some of these issues and represent a more nuanced model. Such a general factor could represent a mental health status that indexes shared variance between psychopathology and wellbeing and thus could better distinguish between, for example, people with low wellbeing/low psychopathology and high wellbeing/low psychopathology. Current general factor models including only psychopathology would not be able to capture this difference. Further, similar to the argument for p factor models better accounting for dynamic symptom changes, such a model would better account for individuals who may cycle between high psychopathology and low wellbeing states, but remain with some form of distress overall. Finally, since wellbeing has been found to additionally distinguish functional outcomes even amongst individuals with mental illness [30], such a general factor could be particularly useful in predicting future functional impairment. In other words, the general factor could be expected to retain the utility of being a parsimonious construct that captures general mental health status and yet be a better predictor of future functional states than a general factor that does not take wellbeing into account. At the same time, the model would retain the distinct aspects of wellbeing and psychopathology, as captured by the specific factors that are free of their shared variance. The model, as a whole, could then be considered to represent a more complete mental health status.

Some previous literature has incorporated wellbeing into general factor models [19,31–33]. Böhnke & Croudace [31] found that Goldberg's General Health Questionnaire (GHQ-12) and the Warwick-Edinburgh Mental Wellbeing Scale (WEMWBS) loaded onto a strong general factor and concluded that the two instruments may not measure distinct constructs. Black et al. [19] used pilot data of the main study the current article uses data from and found a general factor, which they labelled 'internalising distress', that explained the covariance between internalizing psychopathology, externalizing psychopathology and wellbeing. However, their measure of wellbeing was the Child Outcomes Rating Scale, which was originally designed as a self-report outcome measure in child and adolescent therapy and does not strictly measure wellbeing [34]. Finally, St Clair et al. [32] measured various psychopathologies (e.g., depression, generalized anxiety, OCD) as well as self-esteem and wellbeing and found a bifactor model fit well, with a general factor and five specific factors (representing self-confidence, antisocial behaviour, worry, aberrant thought and low mood). They also interpreted the general factor as a construct representing 'distress' and found it was associated with self-reported hazardous behaviour such as substance use and self-harm. However, the researchers allowed cross-loadings and adjusted the model as per modification indices, which may have led to overfitting. Van Erp Taalman Kip and Hutschemaekers [33] fit a model incorporating wellbeing and mental illness with a clinical population and found that while a dual-factor model was supported, the wellbeing factor only explained a small amount of the variance. They concluded that "if independency between the factors is a prerequisite, then only one continuum emerges with psychopathology and wellbeing as bipolar opposites" (p. 1725). However, as noted above, we cannot consider independency between wellbeing and psychopathology to be a prerequisite, as that is likely to be untrue. Further, as the authors concluded, in a clinical sample, the importance of wellbeing may be subsumed by the dominant role of psychopathology. Finally, while these studies have fit models incorporating both psychopathology and wellbeing, to our knowledge, no study has, as yet, evaluated the association of such a general factor to a prospective variable indexing a future functional state.

Functional impairment can be defined as the difficulty in handling the routine demands of everyday life [35] and can be conceptualized as a crucial consequence of mental illness [36,37], albeit it can also be argued to have a bidirectional relationship with mental illness. Empirical research has found that functional impairment is a better predictor of mental health service use than symptoms [38] and is also a valuable measure of treatment effectiveness, in addition to symptom change measures [39,40]. Both objective (e.g., psychiatric hospitalizations) and subjective (e.g., self-report) measures have been used to measure functional impairment. We argue that both have value, as the relationship between objective and subjective measures has been found to be stable, but weak, suggesting that both capture unique aspects of functional impairment [36]. Further, in concert with the recent calls to increase young people's voices in research [41], there is a value-based argument to use self-reports of impairment in research, which highlight the young person's unique perspective on their own impairment. Hence, in this study, we focus on self-reported subjective functional impairment, acknowledging that this may only capture one dimension of functional impairment.

## Aims and objectives

This study aimed to understand whether incorporating wellbeing as another dimension within general factor models of mental health is (a) feasible and (b) useful in predicting future impairment. We refer to the general factor in such a model as $g_{wb}$ to conceptually distinguish it from the p factor.

**Research Question 1.** Can a model with a $g_{wb}$ factor be identified in a school-based adolescent sample?

**Research Question 2.** Is the $g_{wb}$ factor independently and prospectively associated with future impairment?

Based on previous literature [19,32], we predicted that a general factor model with wellbeing would fit the data satisfactorily and that $g_{wb}$ would represent a negative mental state, with negative loadings of the wellbeing items and positive loadings of the psychopathology items. We predicted that $g_{wb}$ would have a significant positive association with future functional impairment, even accounting for the effects of concurrent functional impairment. We also planned to estimate a correlated factor model for comparison purposes and predicted that this model would also have satisfactory fit. Finally, the specific factors in the correlated factors model would also have significant positive (Internalizing, Externalizing) and negative (Wellbeing) associations with future impairment.

## Methodology

We conducted secondary analysis of data collected between 2016/17–2018/19 as part of the HeadStart programme. HeadStart was a 6-year large-scale programme which aimed to explore methods to improve mental health and resilience in young people in England. We included data collected at two time points (Year 7 and Year 9).

## Participants

Overall, the HeadStart programme accessed a school-based sample of 67871 adolescents from six different local authorities in England (Blackpool, Cornwall, Hull, Kent, Newham and Wolverhampton). These six local authorities were identified based on levels of deprivation and engagement with the programme. Hence, the sample might not be representative of all school children in England.

The analytic sample for this study was 15258 adolescents in Year 7 who had participated in Wave 1 and filled out the Strengths and Difficulties Questionnaire and the Short Warwick Edinburgh Mental Wellbeing Scale (defined as responding to at least one question on each measure). As compared to national-level data published by Department for Education in January 2017 [42,43], the study sample at baseline had a higher percentage of children who were White as compared to other ethnicities (study: 74.0%; national: 70.9%, for state-funded secondary schools) and a lower percentage of children with special educational needs (SEN) (study: 11.6%; national: 14.4%, for all school types). The percentage of children ever eligible for Free School Meals (FSM) in our sample was 35.2%. Out of the initial sample, 11535 participants (75.6%) were followed up in Year 9, two years later.

 

## Procedures

Data collection for the study was conducted between January 2017 – June 2021. At the start of the study, parents and carers of children in the longitudinal group were provided with a written information sheet and opt-out consent form, with a deadline of at least two weeks from the date of issue. Opt-outs could be submitted via Freepost, phone, or email to the Data Manager. All children in an eligible year group in participating schools (excepting those whose carers had opted out) were invited to fill out the measures. For each survey session, written child assent was sought and recorded via computer at the beginning of the session. Students filled out measures during their usual school day, in the presence of a teacher or adult. This procedure was approved by the University College London Research Ethics Committee (ID number 7963/003). Further details are available in the HeadStart National Evaluation Report [44]. The data used in the current study were accessed after completion of the study on 30th October 2023 and these data were anonymized, i.e., researchers could not identify participants. This use was covered under the original ethical approval for the study.

## Measures

All measures were based on youth-self-report, except socio-demographic characteristics.

**Socio-demographic characteristics.**  Socio-demographic data, i.e., gender, ethnicity, children with SEN, eligibility for FSM and Index of Deprivation Affecting Children (IDACI) score were obtained through a data linkage with the National Pupil Database. The IDACI score is an area-level deprivation index and measures the proportion of all children aged 0–15 living in income deprived families in a particular area [45].

**Psychopathology.**  The 25-item Strengths and Difficulties Questionnaire (SDQ) was used to measure psychopathology [46,47]. The SDQ contains a total of five subscales: emotional problems, conduct problems, peer problems, hyperactivity and prosocial behaviour. The emotional problems subscale was considered to represent internalizing symptoms and the conduct problems subscale was considered to represent externalizing symptoms, following Patalay et al. [4]. Respondents use a 3-point Likert scale, where higher scores generally indicate higher psychopathology, excepting for one item, which is reverse scored.

**Wellbeing.**  The 7-item Short Warwick Edinburgh Mental Well-being Scale (SWEMWBS) was used to measure subjective wellbeing [48]. Despite appearing to tap psychological and eudaimonic aspects of wellbeing to a greater extent than hedonic wellbeing, the SWEMWBS has been found to be a reliable and unidimensional measure of mental well-being [48]. Respondents use a 5-point Likert scale, where higher scores indicate higher wellbeing and no items are reverse scored.

**Functional impairment.**  In this study, we measured functional impairment using the impact supplement of the SDQ, which measures impairment in social and occupational domains: home life, friendships, classroom learning and leisure activities [38]. Item 1 in the impact supplement was: 'Overall, do you think you have difficulties in one or more of the following areas?'. Individuals who reported no perceived difficulties on Item 1 are considered as having an impact score of zero [49]. For individuals who report any perceived difficulties, the impact score is typically computed using five items. We further *a priori* decided to exclude the 'distress' item (Item 3; which is typically included in the impact score) as this item might be conceptually overlapping with both internalizing difficulties and wellbeing. Hence, we computed impact score by adding Items 4–7, answered on a 4-point Likert scale. Overall, the impact score ranged from 0–12.

## Data analyses

All analyses were conducted using R version 4.3.1 [50].

**Missing data.**  As previously indicated, out of the analysable sample of 15258 at baseline, 11535 participants had been followed up two years later while 3723 had been not. Chi-squared tests and t-tests comparing these two groups on baseline socio-demographic characteristics and the measures of interest were significant (S1 Table), indicating that the sample who were followed up were significantly different on these baseline indicators than the sample who were not

followed up. In particular, the sample who were not followed up were more likely to be boys, those eligible for FSM, those with SEN provision and those with lower wellbeing, higher symptom scores and impairment at baseline. Children from White ethnic groups were also less likely to be followed up as compared to children from Black, Asian and Other ethnic groups, but this was not true when comparing to a Mixed ethnic group. Overall, this is indicative of an attrition bias in the data with at least a Missing at Random (MAR) mechanism, although Missing Not At Random (MNAR) is also possible (see [51] for an explanation of missingness mechanisms). To account for this attrition bias, we included the biased socio-demographic variables as covariates in the analyses and imputed missing data on the initial sample of n = 15258 using multiple imputation by chained equations [52–54].

We used the R package mice version 3.16.0 [55] to conduct multiple imputation under MAR assumption. Even if data are Missing Not At Random, multiple imputation can produce estimates robust against MNAR [56]. For the final dataset entered into the imputation, variable-wise missingness ranged from 0–28.5% and case-wise missingness ranged from 0–92%. All variables (predictors, covariates and outcome variables) included in the analysis were included in the imputation model. All covariates shown to be associated with missingness, as above, were included. Further, we also included all other variables available in the dataset as auxiliary variables in the imputation model, excepting variables excluded on administrative grounds (e.g., participant id) and substantive grounds (e.g., all variables measured post the COVID-19 pandemic). For computational efficiency, we also excluded variables at time point 2 for whom the outbound statistic for imputing future impairment (the main outcome variable) was 0.15 or below [56]. The predictive mean matching method was used for imputation because a) it works well when data do not meet the assumption of multivariate normality b) only plausible values are imputed, allowing distributions to be preserved and c) it is computationally faster. Given available time and complexity of the model, we imputed a total number of m = 20 datasets at 10 iterations and visually checked for convergence, which is suggested to be acceptable [56]. The summary statistics of the imputed datasets are available in S2 Table.

**Structural equation modelling: Measurement model.** To address Research Question 1, we estimated measurement models that tested whether our theorized models fit the data as per pre-defined fit thresholds. The semTools package version 0.5–6.932 in R [57] was used to run models on imputed datasets using the functions *cfa.mi* and *sem.mi* and to pool parameter estimates and standard errors. For the structural models, we converted pooled unstandardized estimates to pooled standardized estimates using a custom function (available in OSF code files), as this functionality was not available in semTools at the time of analyses.

In accordance with literature that generally tests a bifactor and correlated factor model [13], we initially specified two models:

**Bifactor $g_{wb}$ model.** This model was a general factor model using items indexing both psychopathology and wellbeing (Fig 1(a)). It had three specific factors (Internalizing, Externalizing and Wellbeing) and one general factor, which we called the general factor with wellbeing or $g_{wb}$. In concert with recommendations [13], we refer to these specific factors as $g_{wb}$-free specific factors, to conceptually distinguish them from specific factors in a correlated factors model. The specific factors were specified to be orthogonal to each other as the general factor is theoretically supposed to capture the variance shared by the specific factors.

**Correlated factors model.** This model used items indexing both psychopathology and wellbeing, but it did not have a general factor (Fig 1(b)). It was specified to have three specific factors (Internalizing, Externalizing and Wellbeing factors) which were allowed to correlate.

In a post-hoc decision, we also specified the following models:

**Bifactor $g_{wb}$ model with a method factor.** This model was the same as the bifactor $g_{wb}$ model, with the addition of a method factor for all negatively worded items (Fig 1(c)). This model was specified to account for wording effects that can occur when some items are negatively worded (nine out of 10 SDQ items) and others are positively worded (all SWEM-WBS items and one SDQ item). Wording effects could confound the distinctness of the dimensions of psychopathology

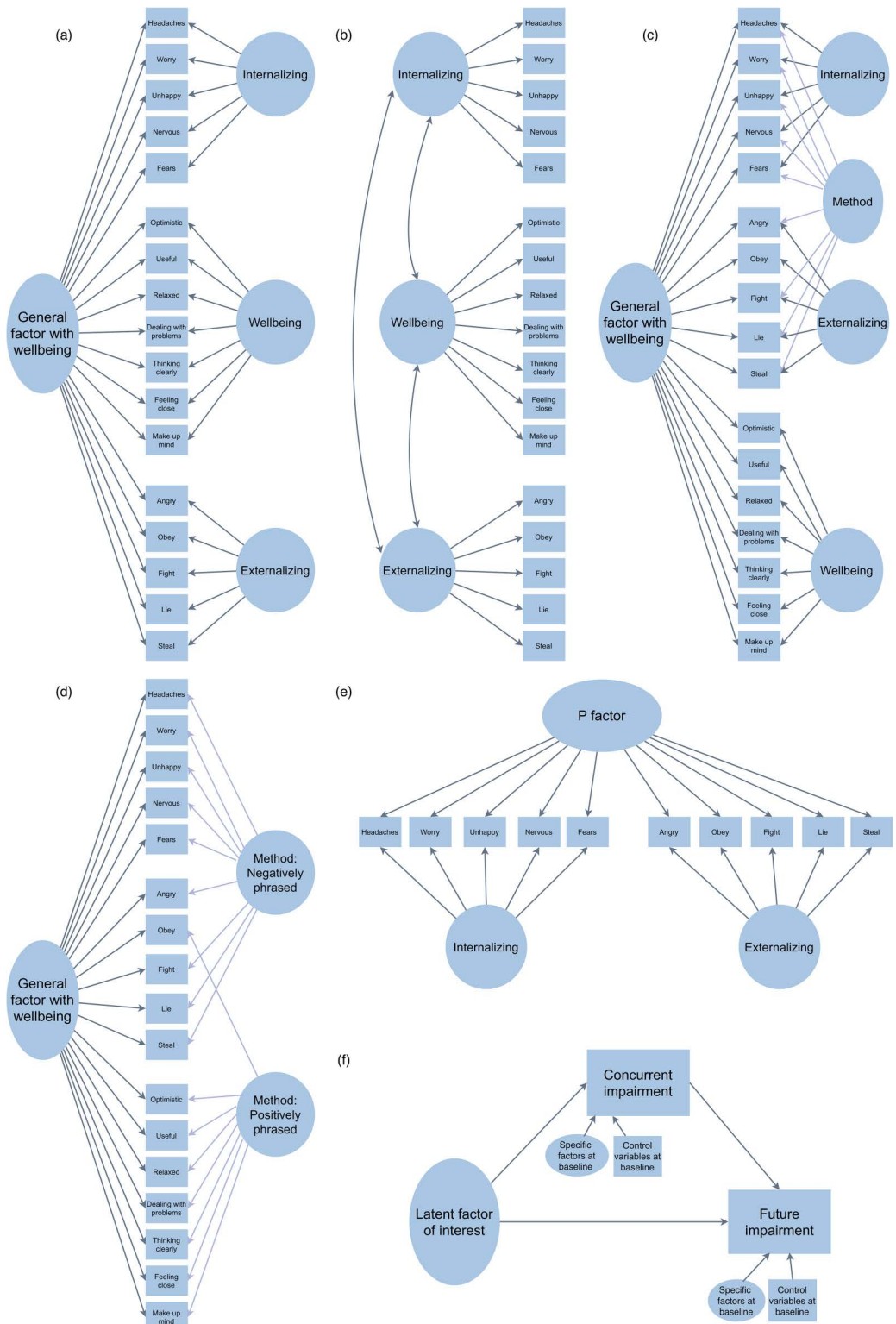

**Fig 1. Measurement models.** (a) Bifactor $g_{wb}$ model (b) Correlated factors model (c) Bifactor $g_{wb}$ with method factor model (d) Bifactor model with method factors (e) Bifactor p model. Structural models. (f) Partial mediation model. For the partial mediation, the latent factor of interest is $g_{wb}$ factor or p factor or specific factors (for correlated factors model), specific factors at baseline are $g_{wb}$-free specific factors or p-free specific factors, depending the measurement model and control variables are gender, ethnicity, FSM, SEN and IDACI score. In all models, rectangles indicate observed variables and ovals indicate latent variables.

and wellbeing. To account for this, we estimated a method factor for all negatively worded items. Loadings of all items on the method factor were constrained to equality to aid identification. When we included a method factor for positively phrased items as well (i.e., all the SWEMWBS items + one item from the SDQ), we found that the variance estimated for this latent factor was negative, despite the model overall having a better fit. Notably, Item 7, a reverse-worded item, is tricky to interpret as positive or negative ('I usually do as I am told') and has shown inconsistent results in previous analyses [58]. Hence, we decided to only include a method factor for negative wording effects in this model, which is in line with previous work on similar questionnaires that has focused on negative wording effects but not positive wording effects [31].

**Bifactor p model.** We also specified a general factor model using items indexing only psychopathology (Fig 1(e)). It had two specific factors (Internalizing, Externalizing) and a general factor, which we called the psychopathology factor, in accordance with the literature on the p factor. This model did not include a method factor since all the items excepting one (SDQ Item 7) were negatively worded. This model was specified for descriptive comparison purposes.

**Bifactor model with two method factors**. This was a general factor model where all the items loaded on the general factor (Fig 1(d)). There were no specific factors for internalizing, externalizing and wellbeing. There were two method factors for positively worded and negatively worded items. This model was specified to understand whether the general factor along with two method factors is a sufficient representation of the underlying structure of the data.

We used the WLSMV estimator in all models, to better represent the categorical nature of the indicator variables. Cluster-robust standard errors with ordinal data are not supported yet in semTools so we could not account for cluster in the analysis model. However, previous studies have shown that this is a small effect [59].

**Structural equation modelling: Measurement and structural models.** Subject to the measurement models having satisfactory fit indices and factor loadings, the corresponding structural model was then simultaneously estimated along with the selected measurement models. The aim of the structural model was to relate the latent factor of interest (the $g_{wb}$ factor, the specific factors from the correlated factors model, the p factor) to future impairment. We included concurrent impairment as a mediator between the latent factor of interest and future impairment, to ensure that we could test for the direct effect of the latent factor on future impairment, accounting for effects due to concurrent impairment. We also included gender, ethnicity, SEN, FSM and IDACI score at baseline as covariates (Fig 1(f)). In the case of general factor models, we also included the $g_{wb}$-free or p-free specific factors from the same model at baseline as covariates.

**Model fit and model comparisons.** All models were checked for fit using the following fit indices thresholds [13,60], which were specified *a priori*: Comparative Fit Index (acceptable fit > 0.90, excellent fit > 0.95); Tucker Lewis Index (acceptable fit > 0.90, excellent fit > 0.95); RMSEA (acceptable fit < 0.08; excellent fit < 0.06); SRMR (fit < 0.08). For the full model with the measurement and structural parts, we manually specified a baseline model for computation of the incremental fit indices, which was the typical independence or null model. Unstandardized estimates were used to test against the null hypothesis. Standardized estimates were used to descriptively compare factor loadings and effect sizes.

For bifactor measurement models, we also calculated bifactor indices such as omega hierarchical and explained common variance [13,61,62]. Omega total ($\omega$) refers to the proportion of variance in an observed total score that can be attributed to all modelled sources of variance (general and specific factors). Omega hierarchical ($\omega_H$) refers to the proportion of variance in an observed total score that can be attributed only to the general factor, partitioning out the specific factors. Omega hierarchical subscale ($\omega_{HS}$) refers to the proportion of variance in an observed subscale score that can be attributed only to the specific factors, controlling for the general factor. Finally, Explained Common Variance represents the proportion of all common variance explained by a factor and ranges from 0 to 1 [13,61].

Our primary objective was to test each model individually for fit. However, we also attempted to compare models as a secondary objective. Where models were nested, they were compared using scaled chi-square difference tests. Most models were non-nested so could not be compared in this manner. Relative fit indices such as AIC and BIC that are typically used to compare non-nested models are not available for the WLSMV estimator. Hence, we descriptively compared non-nested models.

**Sensitivity analysis.** We also estimated the final selected models using the MLR estimator and full-information maximum likelihood to handle missing data. Finally, we also estimated the same models using WLSMV without multiply imputed data.

## Results

### Structural equation modelling: Measurement models

Table 1 depicts the fit indices for all five measurement models estimated. Although all models had satisfactory fit indices as per the more lenient fit thresholds, only Model 3 (bifactor $g_{wb}$ with method) and Model 4 (bifactor p) came close to meeting all the conservative thresholds. The chi-squared test was significant; however this was due to the sensitivity of the chi-squared test statistic to large samples. Table 2 depicts additional bifactor indices. Below, each model's fit indices and factor loadings are summarized. Full results tables are available in S3 Table.

**Bifactor $g_{wb}$ model.** For the bifactor $g_{wb}$ model, the overall model fit was nearly excellent ($\chi2(102, 15258) = 5032.5$, $p < 0.01$, CFI = 0.948, TLI = 0.931, RMSEA = .056, SRMR = 0.049). Standardized factor loadings had a range between 0.25–0.64 for the $g_{wb}$-free internalizing factor, 0.34–0.53 for the $g_{wb}$-free wellbeing factor, 0.41–0.77 for the $g_{wb}$-free externalizing

**Table 1. Measurement Models.**

| Model | Chi-squared | CFI* | TLI * | RMSEA* | SRMR * |
|---|---|---|---|---|---|
| Model 1: Bifactor $g_{wb}$ | p < 0.001 | 0.948 | 0.931 | 0.056 | 0.049 |
| Model 2: Correlated factors | p < 0.001 | 0.922 | 0.908 | 0.065 | 0.064 |
| Model 3: Bifactor $g_{wb}$ with method | p < 0.001 | 0.959 | 0.945 | 0.050 | 0.045 |
| Model 4: Bifactor p | p < 0.001 | 0.971 | 0.949 | 0.054 | 0.042 |
| Model 5: Bifactor with method factors | p < 0.001 | 0.927 | 0.903 | 0.067 | 0.059 |

$g_{wb}$, general factor with wellbeing; p, general psychopathology factor; CFI, Comparative Fit Index; TLI, Tucker-Lewis Index, RMSEA, Root Mean Square Error of Approximation; SRMR, Standardized Root Mean Square Residual.

*Predefined thresholds: CFI & TLI: Acceptable >0.90, Excellent > 0.95; RMSEA: Acceptable < 0.08, Excellent < 0.06; SRMR <0.08.

**Table 2. Additional Bifactor Indices.**

| Model | $\omega_{H/}\,\omega_{HS}$ | ECV | $\omega$ |
|---|---|---|---|
| Model 1: Bifactor $g_{wb}$ | $g_{wb}$: 0.60 | $g_{wb}$: 0.47 | 0.86 |
| | Int: 0.39 | Int: 0.17 | |
| | Ext: 0.40 | Ext: 0.18 | |
| | WB: 0.38 | WB: 0.18 | |
| Model 3: Bifactor $g_{wb}$ with method | $g_{wb}$: 0.68 | $g_{wb}$: 0.48 | 0.86 |
| | Int: 0.34 | Int: 0.14 | |
| | Ext: 0.39 | Ext: 0.17 | |
| | WB: 0.12 | WB: 0.07 | |
| Model 4: Bifactor p | p: 0.47 | p: 0.43 | 0.79 |
| | Int: 0.38 | Int: 0.27 | |
| | Ext: 0.44 | Ext: 0.31 | |
| Model 5: Bifactor with method factors | $g_{wb}$: 0.57 | $g_{wb}$: 0.55 | 0.86 |

$g_{wb}$, general factor with wellbeing; p, general psychopathology factor; INT, internalizing factor; EXT, externalizing factor; WB, wellbeing factor; $\omega_H$, Omega hierarchical; $\omega_{HS}$, Omega hierarchical subscale; ECV, Explained Common Variance; $\omega$, Omega total.

factor and an absolute range of 0.27–0.68 for the complete mental state factor. The mean of the absolute standardized factor loadings was moderate for all latent factors (0.48 for the $g_{wb}$-free internalizing factor, 0.51 for the $g_{wb}$-free externalizing factor, 0.44 for $g_{wb}$-free wellbeing factor and 0.44 for the $g_{wb}$ factor). Although some of these loadings are relatively low, low factor loadings are common with child and adolescent populations (e.g., Patalay et al., 2015; Black et al., 2019). Highest standardized loadings on the $g_{wb}$ factor included 'I am often unhappy, downhearted and tearful' (0.68), 'I've been thinking clearly' (−0.62), 'I've been dealing with problems well' (−0.56), 'I've been feeling relaxed' (−0.55) and 'I get very angry and often lost my temper' (0.54).

Following Caspi et al. [13], the omega hierarchical for $g_{wb}$ was acceptable (0.60) and suggested that a total score of SDQ and SWEMWBS items would predominantly reflect individual differences on the general factor. Following Stucky and Edelen's [62] guidelines, the proportion of common variance explained by the general factor in this model (47%) was not high enough to warrant a unidimensional model.

**Correlated factors model.** For the correlated factors model, the overall model fit was not excellent, but acceptable ($\chi2(116, 15258) = 7566.8$, $p < 0.01$, CFI = .922, TLI = 0.908, RMSEA = .065, SRMR = 0.064). Standardized factor loadings had a range between 0.54–0.85 for the internalizing factor, 0.48–0.77 for the wellbeing factor and 0.51–0.78 for the externalizing factor. The mean standardized loadings were fairly high (0.66 for the internalizing factor, 0.63 for the externalizing factor and 0.63 for the wellbeing factor). Further, the covariance was 0.44 between internalizing and externalizing factors, −0.52 between internalizing and wellbeing factors and −0.48 between externalizing and wellbeing factors.

**Bifactor $g_{wb}$ with method factor.** For the bifactor $g_{wb}$ model with the method factor, the overall model fit was excellent ($\chi2(101, 15258) = 3994.2$, $p < 0.01$, CFI = 0.959, TLI = 0.945, RMSEA = 0.050, SRMR = 0.045). Standardized factor loadings had slightly reduced for the $g_{wb}$-free internalizing factor (Mean: 0.45) and the $g_{wb}$-free externalizing factor (Mean: 0.50) as compared to the original bifactor model. Standardized loadings on the $g_{wb}$-free wellbeing factor were low in this model (Mean: 0.25), with four items, in particular, having low factor loadings that albeit were significantly different from zero ("feeling relaxed", "dealing with problems well", "thinking clearly", "been able to make up my own mind"). The mean standardized loading on the $g_{wb}$ factor was similar (Mean: 0.44) although the contributions of the internalizing and externalizing items had reduced, presumably due to some of their variance now being accounted for by the method factor. In turn, the contributions of wellbeing items had increased, especially those items that now had low loadings on the $g_{wb}$-free and method-free wellbeing specific factor. In terms of items, highest standardized loadings were similar, including 'I've been thinking clearly' (−0.62), 'I've been dealing with problems well' (−0.56), 'I've been feeling relaxed' (−0.55), 'I am often unhappy, downhearted and tearful' (0.54) and 'I've been feeling useful' (−0.54). Notably, although 'I get very angry and often lose my temper' still had the highest loading on $g_{wb}$ out of the externalizing items, its standardized loading had reduced from 0.54 to 0.41.

The omega hierarchical for $g_{wb}$ was acceptable (0.68) and suggested that a total score of SDQ and SWEMWBS items would predominantly reflect individual differences on the general factor. For the specific factors, the omegas hierarchical subscale for Int, Ext and WB were 0.34, 0.39 and 0.12 respectively, suggesting that the subscale total scores would reflect the general factor more than the intended factor, particularly for the Wellbeing subscale. The proportion of common variance explained by the general factor in this model (48%) was also not high enough to warrant a unidimensional model [62].

**Bifactor p.** As per the fit indices, this model had excellent fit ($\chi2(25, 15258) = 1154.1$, $p < 0.01$, CFI = 0.971, TLI = 0.949, RMSEA = .054, SRMR = 0.042). Standardized factor loadings had a range of 0.17–0.64 for the internalizing factor (Mean: 0.47), 0.40–0.69 for the externalizing factor (Mean: 0.53) and 0.31–0.72 for the p factor (Mean: 0.42). For the first time in all models, one of the factor loadings on the general factor (in this case, p factor) was not significant and had an estimate of 0.08 (Item 7: I usually do as I am told). Highest standardized loadings on the p factor included 'I am often unhappy, downhearted or tearful' (0.72), 'I get very angry and often lose my temper' (0.55) and 'I get a lot of headaches, stomachaches or sickness' (0.54).

In this model, the omega hierarchical for the general factor, p, was lower than usually considered acceptable [13], suggesting that the variance in the total score of all items explained by the p factor was moderate and the specific factors were also important in explaining variance in the observed total score. Similarly, the proportion of common variance explained by the general factor in this model (43%) was also not high enough to warrant a unidimensional model [62].

**Bifactor with method.** This model was estimated to test the hypothesis that a general factor along with two method factors was sufficient to explain the data. The model had acceptable fit as per lenient fit indices ($\chi^2$(102, 15258) = 7031.8, $p < 0.01$, CFI = 0.927, TLI = 0.903, RMSEA = .067, SRMR = 0.059). The standardized loadings on the general factor $g_{wb}$ ranged between 0.21–0.76 (Mean: 0.46). However, the factor loadings for the negative wording factor clearly split along the internalizing items (all positive loadings) and externalizing items (all negative loadings) of the SDQ, suggesting that it is not representing a negative wording factor, as we would expect such a factor to have loadings from all the specified items in the same direction. Similarly, for the positive wording factor, the factor loadings split between the one SDQ item (positive loading) and all the SWEMWBS items (all negative loadings), again suggesting that it is not representative of a positive wording factor. Hence, we considered this a misspecified model based on factor loadings.

Finally, based on our reviewer's suggestion, we also fit an additional pure unidimensional model with a general factor ($g_{wb}$) on which all items loaded. There were no specific factors or method factors. This model was fit on an archived version of the dataset, since we no longer had access to the original dataset. This model showed poor fit ($\chi^2$(119, 15255) = 23258.556, $p < 0.01$, CFI = 0.758, TLI = 0.723, RMSEA = 0.113, SRMR = 0.116), suggesting that a single general factor is not adequate to explain the variance in the data. Further information as well as the explanation of minor differences between the original and archived dataset are reported in S7 Text.

Since Model 1 (bifactor $g_{wb}$) and Model 3 (bifactor $g_{wb}$ with method factor) were nested, they could be compared using a scaled chi-square difference test. This indicated that Model 3 (bifactor $g_{wb}$ with method factor) was superior ($\Delta\chi^2 = 500$, df = 1, $p < 0.001$). Since the method factor in the bifactor model included 9/10 SDQ items, it is difficult to conclusively establish that it truly represents the method effect of negative-phrased items only. It could also be capturing elements of shared psychopathology that are non-specific to internalizing and externalizing but that are not shared with wellbeing. However, constraining the loadings of items on the method factor to equality should work against this.

Hence, for the next stage of the analysis, we carried forward the correlated factors model, the bifactor $g_{wb}$ with a method factor model and bifactor p model.

## Structural equation modelling: Measurement and structural models

For all three selected models, we specified a partial mediation model along with the original measurement model. Table 3 depicts the fit indices for these structural models.

As seen in Table 3, only the bifactor $g_{wb}$ mediation model with the method factor and the adjusted correlated factors model met the pre-defined fit thresholds. None of the models met the more conservative criteria for fit indices. Tables 4 and 5 contain the regression coefficient estimates and the estimates for direct and indirect paths for the bifactor $g_{wb}$ mediation model and the adjusted correlated factors model. Below, key findings from each model are summarized.

**Table 3. Structural Models.**

| Model | Chi-squared | CFI* | TLI* | RMSEA* | SRMR* |
|---|---|---|---|---|---|
| Bifactor $g_{wb}$ with method mediation model | p<0.001 | 0.939 | 0.925 | 0.042 | 0.039 |
| Adjusted correlated factors mediation model | p<0.001 | 0.913 | 0.900 | 0.048 | 0.056 |
| Bifactor p mediation model | p<0.001 | 0.915 | 0.884 | 0.050 | 0.036 |

$g_{wb}$, general factor with wellbeing; p, general psychopathology factor; CFI, Comparative Fit Index; TLI, Tucker-Lewis Index, RMSEA, Root Mean Square Error of Approximation; SRMR, Standardized Root Mean Square Residual.

*Predefined thresholds: CFI & TLI: Acceptable >0.90, Excellent >0.95; RMSEA: Acceptable <0.08, Excellent <0.06; SRMR <0.08.

**Table 4. Parameter estimates from structural equation models.**

| Variables | Pooled standardized estimate | Pooled standardized standard error |
|---|---|---|
| **Bifactor $g_{wb}$ with method mediation model** | | |
| Future imp~$g_{wb}$ | 0.31* | 0.019 |
| Future imp ~concurrent imp | 0.12* | 0.017 |
| Concurrent imp~$g_{wb}$ | 0.59* | 0.010 |
| Future imp~$g_{wb}$-free and method-free INT | 0.11* | 0.013 |
| Concurrent imp~$g_{wb}$-free and method-free INT | 0.26* | 0.014 |
| Future imp~$g_{wb}$-free and method-free EXT | 0.059* | 0.014 |
| Concurrent imp~$g_{wb}$-free and method-free EXT | 0.15* | 0.013 |
| Future imp~$g_{wb}$-free and method-free WB | 0.058* | 0.020 |
| Concurrent imp~$g_{wb}$-free and method-free WB | 0.14* | 0.016 |
| **Adjusted correlated factors mediation model** | | |
| Future imp~INT | 0.14* | 0.013 |
| Concurrent imp~INT | 0.36* | 0.010 |
| Future imp~EXT | 0.07* | 0.013 |
| Concurrent imp~EXT | 0.21* | 0.011 |
| Future imp~WB | −0.08* | 0.012 |
| Concurrent imp~WB | −0.14* | 0.010 |
| Future imp~concurrent imp | 0.21* | 0.011 |

Imp, impairment; $g_{wb}$, general factor with wellbeing; INT, internalizing factor; EXT, externalizing factor; WB, wellbeing factor.

*$p < 0.05$

**Table 5. Parameter estimates for direct and indirect effects.**

| Model | Effect | Pooled standardized estimate | Pooled standardized standard error |
|---|---|---|---|
| Adjusted correlated factors mediation model | Direct (INT) | 0.14* | 0.013 |
| | Indirect (INT) | 0.08* | 0.004 |
| | Direct (EXT) | 0.07* | 0.013 |
| | Indirect (EXT) | 0.04* | 0.003 |
| | Direct (WB) | −0.08* | 0.012 |
| | Indirect (WB) | −0.03* | 0.003 |
| Bifactor $g_{wb}$ mediation model with method factor | Direct | 0.31* | 0.019 |
| | Indirect | 0.07* | 0.010 |

$g_{wb}$, general factor with wellbeing; INT, internalizing factor; EXT, externalizing factor; WB, wellbeing factor.

*$p < 0.05$.

**Bifactor $g_{wb}$ mediation model with method factor.** The overall model fit of the bifactor $g_{wb}$ partial mediation model with the method factor was satisfactory ($\chi2(263, 15258) = 7210.0$, $p < 0.01$, CFI = 0.939, TLI = 0.925, RMSEA = 0.042, SRMR = 0.039). The estimate for the direct path from the general factor $g_{wb}$ to future impairment was positive and significant (unstandardized estimate: 2.51, $p < 0.001$; pooled standardized estimate: 0.31). The estimate for the indirect path from the general factor to future impairment via concurrent impairment was also positive and significant, though lower in terms of effect size (unstandardized estimate: 0.59, $p < 0.001$; pooled standardized estimate: 0.07). With respect to the $g_{wb}$-free and method-free specific factors, both the Int and Ext factors were significantly and positively associated with

concurrent impairment (Int: pooled standardized estimate: 0.26; Ext: pooled standardized estimate = 0.15), however the relationship with impairment became weaker over time (Int: pooled standardized estimate: 0.11; Ext: pooled standardized estimate = 0.059). The $g_{wb}$-free and method-free Wellbeing factor had small but significant positive associations with impairment in this model. With respect to the control variables, SEN status and FSM status were positively and significantly associated with concurrent impairment, with the relationship maintained over time. Gender and ethnicity were associated with concurrent impairment, with males (as compared to females) reporting lower impairment and Asian and Black adolescents (as compared to white adolescents) reporting lower impairment. The relationship of gender and ethnicity with impairment became stronger over time. IDACI score was significantly associated with concurrent impairment, but the relationship became non-significant over time (See S4 Table for full results of control variables).

**Adjusted Correlated Factors Mediation Model.** This model simultaneously included the internalizing, externalizing and wellbeing factors. Thus, it can be considered an adjusted correlated factors model as it estimates the unique effect of each specific factor, adjusting for the effect of the other specific factors. The overall model fit was satisfactory, but descriptively worse as compared to the bifactor $g_{wb}$ mediation model ($\chi 2$ (280, 15258) = 10146.3, p < 0.01, CFI = 0.913, TLI = 0.900, RMSEA = 0.048 and SRMR = 0.056). The internalizing and externalizing factors had positive and significant associations with concurrent impairment (internalizing: unstandardized estimate: 1.82, p < 0.001, pooled standardized estimate: 0.36; externalizing: unstandardized estimate: 0.72, p < 0.001, pooled standardized estimate: 0.21), while the wellbeing factor had a negative and significant association with concurrent impairment (unstandardized estimate: −0.81, p < 0.001, pooled standardized estimate: −0.14). Hence, the strongest association was with the internalizing factor and the weakest association with the wellbeing factor. All associations with impairment decreased over time, with the internalizing factor maintaining the relatively strongest association (unstandardized estimate: 0.76, p < 0.001, pooled standardized estimate: 0.14), followed by the externalizing factor (unstandardized estimate: 0.28, p < 0.001, pooled standardized estimate: 0.075) and then the wellbeing factor (unstandardized estimate: −0.50, p < 0.001, pooled standardized estimate: −0.08). The pattern of associations with the control variables remained similar to the bifactor $g_{wb}$ mediation model (See S4 Table for full results). We also estimated separate unadjusted models for the internalizing factor, externalizing factor and wellbeing factor as the latent factor of interest associated with impairment; none of these models met the pre-defined fit thresholds (S5 Table).

**Bifactor p mediation model.** The overall model fit of the bifactor p partial mediation model was adequate as per some fit criteria but did not meet all fit thresholds ($\chi 2$(119, 15258) = 4567.9, p < 0.01, CFI = .915, TLI = 0.884, RMSEA = .050, SRMR = 0.036). Hence, we decided not to interpret path coefficients from this model.

The bifactor $g_{wb}$ mediation model satisfactorily fit the data whereas the correlated factors mediation model just about fit the data and the bifactor p model failed to meet all required fit indices. Owing to the non-nested nature of the bifactor and correlated factor structural models, we could not statistically compare them. Hence, we considered the bifactor $g_{wb}$ mediation model as representing the best model fit to our data, with the correlated factors mediation model as also viable.

### Sensitivity analysis

To test the robustness of the results, we also estimated the bifactor $g_{wb}$ with method model, the bifactor p model and the correlated factors model with the MLR estimator with full information maximum likelihood to account for missing data (S6 Table A and B). The overall pattern of results was similar, but the incremental fit indices were lower for all models (bifactor $g_{wb}$ with method: CFI: 0.907; TLI: 0.886; bifactor p: CFI: 0.894; TLI: 0.856; correlated factors: CFI: 0.871; TLI: 0.852). The fit for just the measurement models were very good for both bifactor models (bifactor $g_{wb}$ with method: CFI: 0.950; TLI: 0.932; bifactor p: CFI: 0.958; TLI: 0.924) but not for the correlated factors model (CFI: 0.906; TLI: 0.890). With the WLSMV estimator without multiple imputation (Cases used: 8709), the results were similar to the current results (S6 Table C). Finally, we tested for the analytic decision of carrying forward the bifactor $g_{wb}$ model with the method factor for the structural model; in case we had carried forward the bifactor $g_{wb}$ model without the method factor instead, it would still

have met the prespecified fit indices at the structural model stage (CFI: 0.934; TLI: 0.919; RMSEA: 0.043; SRMR: 0.043), the associations with impairment for the general factor were similar and the broad conclusions would have remained unchanged (S6 Table D).

## Discussion

This study aimed to test whether a general factor model with a wellbeing dimension is feasible and useful. Our first research question related to feasibility, i.e., whether such a general factor $g_{wb}$ could be identified in a school-based adolescent sample. To this end, we estimated various latent variable models incorporating psychopathology and wellbeing dimensions.

We found that most models had acceptable, but not excellent fit to the data, namely, the bifactor $g_{wb}$ model, the bifactor $g_{wb}$ with method model, the bifactor p model and the correlated factors model. Notably, while the correlated factors model had acceptable fit as per our pre-defined fit indices, the fit indices were descriptively lower than the other models. Further, in a direct model comparison test, the bifactor $g_{wb}$ with method model was superior to the bifactor $g_{wb}$ model. Overall, this indicates that, for representing the overall structure of mental health, the bifactor $g_{wb}$ model with the method factor, the bifactor p model and the correlated factors model are all viable representations of the underlying data structure, with a tentative preference for the bifactor models, which is similar to what has been found previously with child and adolescent samples [4,14–16]. Neither the bifactor model with the method factors only or the pure unidimensional model (the latter fit on the archived dataset) were viable models, suggesting that a single general factor does not adequately represent the structure of mental health. This was also corroborated by the explained common variance in all bifactor models, which suggested that unidimensionality was not warranted. This is consistent with previous findings about unidimensional models in the literature [4,14–16]. Overall, this suggests that there are both shared and distinct aspects between the internalizing, wellbeing and externalizing dimensions; if there was nothing shared, the general factor models would not fit and if there was nothing distinct, the unidimensional model would fit adequately. This is further supported by the finding that there were moderate associations between the specific factors in the correlated factors model. While moderate associations between the internalizing and externalizing are well-established, it is interesting that wellbeing too had *moderate* (negative) associations with the internalizing and externalizing factor, especially the internalizing factor, which it has previously been suggested to conceptually overlap with to a greater extent [19].

The shared underlying construct ($g_{wb}$) between (lack of) wellbeing and psychopathology appears to be a negative mental state. The mean standardized absolute factor loadings on the general factor for the bifactor $g_{wb}$ with method model was 0.44, which was similar to the mean loading on the general factor for the bifactor p model (0.42). The items with highest loadings on the $g_{wb}$ factor were 'I am unhappy, downhearted and tearful', 'I get very angry and often lose my temper', 'I've been thinking clearly', 'I've been dealing with problems well' and 'I've been feeling relaxed'. Of note, most studies have found such a general factor to represent poor rather than good mental states [19,31,32]; it could be that when measuring a construct such as wellbeing, it is easier to find a shared commonality on lack of wellbeing whereas positive wellbeing states might be more multidimensional [31]. Such a shared commonality between (lack of) wellbeing and psychopathology could represent more substantive, higher-order tendencies for poorer psychological wellbeing and higher psychopathology, especially since there are shared genetic influences between mental health and mental illness [21]. For example, this general factor could be capturing a construct such as self-directedness from Cloninger's psychobiological model of personality [63,64]. Self-directedness is conceptualized as a self-regulatory domain of human personality, with a heritable component, and has been linked to both different aspects of wellbeing (e.g., positive affect, life satisfaction; [63]) as well as psychopathology (e.g., emotional and behaviour problems; [65]). However, we also note that substantive interpretations of general factors have been subject to much debate and general factors could also be indexing current mental health status (e.g., distress or impairment), rather than representing antecedents of wellbeing and illness [11].

Comparing standardized factor loadings on the specific factors in the correlated factors model to the $g_{wb}$-free specific factors in the bifactor $g_{wb}$ with method model could suggest which items are more indicative of the general factor and the specific factors [1]. With respect to internalizing items, the factor loadings of "headaches" and "unhappy" substantially reduced from the correlated-factors to the bifactor $g_{wb}$ model, indicating that these items were more indicative of the general factor, whereas the factor loadings for "worry" and "fears" did not change much, indicating that these items formed an internalizing-specific factor even after the general factor has been partialled out. With respect to externalizing items, the factor loadings of "angry" and "lie" reduced substantially from the correlated factors to the bifactor model, while the factor loading of "steal" reduced moderately, the factor loading of "obedient" remained similar and the factor loading of "fight" on the specific factor actually increased in the bifactor model. This indicates that "steal", "obedient" and "fight" might have been capturing quite specific dimensions of externalizing whereas "angry" and "lie" might have been more indicative of the general factor. With respect to the wellbeing items, the factor loadings for "feeling relaxed" "thinking clearly" "dealing with problems" and "make up mind" reduced substantially from the correlated factors to the bifactor model, indicating that these items were more indicative of the general factor, while the loadings for "feeling useful" and "feeling close" reduced moderately and "optimistic" remained similar. Overall, for this dataset, the $g_{wb}$-free and method-free INT factor seems to be capturing aspects of anxiety and fear, whereas the $g_{wb}$-free and method-free EXT factor seems to be capturing aspects of antisocial behaviour. The $g_{wb}$-free and method-free WB factor is more difficult to interpret and less robust due to its low factor loadings; however, it is notable that all wellbeing items loading on the general factor seem to capture internal aspects of lack of wellbeing whereas "feeling close" is the only item that implies more external aspects such as relationships. Emotional states of low mood, anger, (lack of) feeling useful, (lack of) feeling relaxed, (lack of) thinking clearly and so on appear to be more transdiagnostic as they are more shared between internalizing, externalizing and wellbeing dimensions rather than remaining within the specific factors. While the current study has limitations in terms of the specific measures used, it provides a proof of concept that representing the relationship between (lack of) wellbeing and psychopathology in the form of a model including both general and specific factors allows one to acknowledge both the shared and distinct aspects of (lack of) wellbeing with psychopathology [29]. One could imagine a scenario wherein estimated factor scores on such a general factor and the $g_{wb}$-free specific factors can together be used to more accurately and comprehensively describe the mental health status of individuals. In such a hypothetical scenario, an adolescent's mental health status could be described using multiple dimensions. Their score on the general factor could be a parsimonious representation of their general (negative) mental state, which would usefully incorporate lack of wellbeing in addition to psychopathology. Additionally, their score on the internalizing factor could be representative of their fear/anxiety state, their score on the externalizing factor representative of their antisocial behaviour and their score on the wellbeing factor representative of the extent to which they report distinct wellbeing states. While the specific aspects of such a model will certainly need to be refined—for example, the consideration of other dimensions beyond internalizing and externalizing would be very reasonable, based on previous studies having done the same [14,18]—such a dimensional representation of a "complete mental state" might have advantages over more discrete representations such as the quadrant model, which inherently force the individual to be in one of a limited number of categories [19,29]. Notably, while factor scores provide model-based estimates of the latent dimensions, our bifactor reliability indices are more informative regarding the common practice of using unit-weighted total scores. In our models, these indices suggested that simple total scores across all items would predominantly reflect the general factor and subscale totals would also primarily reflect the general factor, with some unique variance from the specific factor. This suggests that factor scores might allow for a cleaner distinction between general and specific factors, whereas unit-weighted total scores may be less interpretable as proxies for their respective specific factors. Hence, using structural equation modelling can be useful in future research (a) to understand the complex relationship between wellbeing and psychopathology and (b) to distinguish between common and specific sources of variance in a cleaner manner.

Our second research question related to the utility of a general factor involving both psychopathology and wellbeing dimensions in independently and positively predicting future impairment. To do this, we estimated structural models with the latent factor of interest as the predictor variable and concurrent and future impairment as mediator and outcome variables.

We found that on this dataset, a general factor model incorporating psychopathology, wellbeing and impairment fit better than a general factor model with just psychopathology and impairment. Specifically, the bifactor $g_{wb}$ partial mediation model, which predicted future impairment from the $g_{wb}$ factor via concurrent impairment had good fit indices. The adjusted correlated factors partial mediation model, which predicted future impairment from the internalizing, externalizing and wellbeing factors via concurrent impairment also had satisfactory fit indices. However, the bifactor p mediation model, which predicted future impairment from the p factor via concurrent impairment, did not meet all pre-defined fit thresholds. Overall, this suggests that a 'complete mental state' model which includes the dimensions of internalizing, externalizing, wellbeing as well as their shared aspects, when associated with concurrent and future impairment, represents a more satisfactory fit to the underlying data structure, than a model that just incorporates the dimensions of internalizing, externalizing and their shared aspects. Note that since the bifactor p mediation model failed to meet criteria by only a small margin, we cannot outright reject this as a viable model, but consider it to be worse-performing on this dataset than the bifactor $g_{wb}$ mediation model.

Further, we found that the $g_{wb}$ factor is significantly and positively associated with future impairment measured two years later. For every 1 standard deviation increase in $g_{wb}$ scores at time 1, impairment scores two years later tended to be higher by 0.31 standard deviations. The direct path from $g_{wb}$ to future impairment was stronger than the indirect path, which had small effect sizes, suggesting that $g_{wb}$ is independently associated with future impairment, even after accounting for concurrent impairment. This finding is aligned with the literature that has found wellbeing to be associated with poor and good functioning [20,28,30]. What is interesting that the shared underlying dimension between (lack of) wellbeing and psychopathology has the strongest association with future functioning as compared to any of the specific factors. This is suggestive of the potential utility of incorporating both wellbeing and psychopathology into measurements of general mental states to inform identification of those at risk for future negative outcomes. Unexpectedly, the weakest path across all models was the path from concurrent impairment to future impairment, albeit it was still positive and significant. This suggests that the general factor might account for most of the expected relationship between concurrent and future impairment. Although we did not change our model to keep with our initial goal of confirmatory analyses, future research might consider using other models than mediation models to represent the relationships between these variables.

Beyond the general factor, even the $g_{wb}$-free Int and Ext factors may be useful in predicting future impairment. These specific factors seem to have retained significant, albeit low, associations with impairment, even after partialling out the general factor. This is conceptually similar to the adjusted correlated factor model, which estimates the unique effect of each specific factor adjusting for the other factors. On the other hand, unadjusted models for these specific factors did not have good fit, suggesting that only using one of the specific factors might not be adequate in predicting future impairment. However, there is more of a caveat to interpreting the associations of the $g_{wb}$-free specific factors with impairment; for the $g_{wb}$-free Ext/WB factors, the effect sizes were low, with relatively higher standard errors, suggesting lower confidence in these associations; this is similar to what Caspi et al. [13] also concluded regarding the stability and interpretation of the specific factors.

Overall, our study tentatively supports the feasibility of a 'complete mental state' model which includes a general factor representing the shared underlying dimension between psychopathology and (lack of) wellbeing as well as specific factors representing internalizing, externalizing and wellbeing specific dimensions. In terms of implications for research, our study supports previous calls that argue against a simplistic conceptualization of psychopathology and wellbeing as totally unidimensional or totally independent and adds to the evidence that that mental illness and wellbeing have both shared and distinct dimensions [19,22,29,32]. A key area for future research might be to attain conceptual and empirical

agreement on what the shared and distinct dimensions capture. Notably, in our model, the wellbeing-specific factor was the most difficult to interpret and least robust in terms of its associations with future impairment; this could be because of the limitations of the comprehensiveness of the wellbeing measure used, which has been found to be unidimensional [66]. One priority for the future may be to identify what, if any, are the distinct aspects of wellbeing that are *not* shared with psychopathology. These could be aspects such as having positive relationships, finding meaning and experiencing positive affect, which are typically not captured by measures of psychopathology but are part of wellbeing models [64,67].

In terms of implications for policy and practice, our study provides some empirical evidence to support calls to measure wellbeing in addition to mental health symptoms when measuring mental health status [68,69], especially for school-based non-clinical samples, as the incorporation of wellbeing in the model seemed to lead to best fit and aided in predicting future impairment. Previous literature has identified the unique contribution of wellbeing and quality of life to predicting protective factors, outcomes and referrals to CAMHS services in the UK, over and above mental health symptoms [27,28,70]. Considering that CAMHS services are often overloaded with referrals, leading to a prioritization of adolescents in crisis [71], measuring both wellbeing and mental health symptoms and conducting an integrated assessment of mental health status based on these measures might prove particularly informative in screening for targeted preventative interventions to prevent young people from reaching such crisis points. Such a measurement is likely to be more inclusive and capture more variability in mental health status [72], which could be particularly useful for school and community samples, where the goal is early identification. The longitudinal nature of our study is a strength here, as it suggests the importance of wellbeing to predicting functional outcomes over time. Secondly, while recognizing that brief self-report measures are probably the most pragmatically useful measures of mental health status in such settings, our study also tentatively argues for going beyond sum score approaches to synthesizing the information from these self-report measures, especially when both mental illness and wellbeing measures are used. Even if existing validated measures are drawn upon for purposes of pragmatics and consistency, there can still be rich information available about the transdiagnostic and distinct positive and negative dimensions of mental health states. Further, different dimensions might be differentially predictive of outcomes (e.g., the externalizing dimension in our study seemed to particularly capture "fighting" and "stealing" and not just "anger", which could be a useful marker of antisocial behaviour).

Our study has the following limitations. Firstly, in the case of all structural equation modelling, what one gets is what one puts in. Our results are necessarily a product of the measures we had, which in this case were limited to the SDQ and the SWEMWBS. Hence, we do not claim to have the most accurate representation of the wellbeing and psychopathology dimensions but rather aim to provide a proof-of-concept of how these dimensions are likely related. Using more comprehensive measures of psychopathology and wellbeing would be important to determine the most appropriate number of specific factors in the model as well as to flesh out what the wellbeing-specific dimension would represent. Secondly, both our predictor variable and outcome variable had the same informant (the adolescents themselves), leading to a risk of common method variance. The fact that the variables in question were measured two years apart somewhat alleviates this, albeit our mediator variable was measured at the same time as the predictor. Finally, it would be remiss to not acknowledge the growing criticism of bifactor models and overfitting with large sample sizes [12], which represents a risk in our sample as well. However, as Caspi et al. [13] argue, the statistical approach taken to general factor modelling (S-1 vs bifactor vs other models) does not seem to influence findings. The more relevant argument is the conceptual argument around the substantive validity and utility of a general factor itself. Here, we do not make any claims about etiological substantiveness of a general factor but use it to conceptualize a model representing the shared and distinct aspects of psychopathology and wellbeing. By doing so, we argue that it is not enough to merely include wellbeing as an additional measure when measuring mental health status in adolescents, but there also needs to be a deeper consideration of how the dimensions of psychopathology and wellbeing are related and can be modelled to provide a more accurate, comprehensive and useful picture of a young person's mental health status.

## Supporting information

**S1 Table. Missingness by variables of interest at baseline.**
(DOCX)

**S2 Table. Results of multiple imputation for variables of interest.**
(DOCX)

**S3 Table. Factor loadings for measurement models.**
(DOCX)

**S4 Table. Factor loadings and parameter estimates for structural models.**
(DOCX)

**S5 Table. Unadjusted models with specific factors.**
(DOCX)

**S6 Table. Sensitivity analysis.**
(DOCX)

**S7 Text. Additional analyses.**
(DOCX)

## Acknowledgments

The authors wish to thank Jan Boehnke for providing comments on an earlier version of the manuscript.

## Author contributions

**Conceptualization:** Ritika Chokhani, Suzet Tanya Lereya, Jessica Deighton.

**Data curation:** Ritika Chokhani.

**Formal analysis:** Ritika Chokhani.

**Supervision:** Suzet Tanya Lereya, Jessica Deighton.

**Writing – original draft:** Ritika Chokhani.

**Writing – review & editing:** Ritika Chokhani, Suzet Tanya Lereya, Jessica Deighton.

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
