## [Decision Letter · Decision Letter 0]

29 Jul 2025

PONE-D-25-20026Incorporating wellbeing into general factor models: a more complete mental state?PLOS ONE?

Thank you for submitting your manuscript to PLOS ONE. After careful consideration, we feel that it has merit but does not fully meet PLOS ONE’s publication criteria as it currently stands. Therefore, we invite you to submit a revised version of the manuscript that addresses the points raised during the review process.

We look forward to receiving your revised manuscript.

Kind regards,

Zheng Zhang

Academic Editor

PLOS ONE

Additional Editor Comments:

This study has a large sample size, employs innovative methods, and presents a well-structured introduction. The data provided in the supplementary materials are also robust. However, I recommend standardizing the format of the p-values to avoid using scientific notation. In addition, I suggest expanding the discussion section to include more practical implications of the methods used, which would help other researchers better understand and apply them.

Reviewers' comments:

Reviewer's Responses to Questions

**Comments to the Author**

1. Is the manuscript technically sound, and do the data support the conclusions?

Reviewer #1: Yes

2. Has the statistical analysis been performed appropriately and rigorously?

Reviewer #1: Yes

3. Have the authors made all data underlying the findings in their manuscript fully available?

Reviewer #1: No

4. Is the manuscript presented in an intelligible fashion and written in standard English?

Reviewer #1: Yes

Reviewer #1: Thank you for the opportunity to review the article “Incorporating wellbeing into general factor models: a more complete mental state?”. The paper is thorough, well presented, and addresses an interesting topic within the literature, namely whether global mental health status might be modelled more accurately through the inclusion of wellbeing alongside psychopathology. The very large sample size, recruited across schools in six different regions of the UK, suggests sufficient statistical power to test the proposed measurement and structural models. My few comments and suggestions for improvement mostly relate to methodology or the results. It is my opinion that the article could appropriate for publication in PlOSONE if these minor issues are addressed:

The method for dealing with missing data was convincing. However, as a small detail, it was unclear to me from the text how the models were fit using the multiple imputed datasets. Was this using the lavaan.mi function? I think readers would benefit if this final part of the imputation process (e.g. pooling) was also explained.

It would be of interest for readers to know the unidimensional internal consistency reliability (eg., ω or α) of the measures, with confidence intervals, in the study sample.

The paper tested several competing measurement models that included indicators of psychopathology and wellbeing. A “bifactor-p” model without wellbeing items was tested as a comparison. The last model tested was labelled as “Unidimensional with method”, which I found misleading, because this is still a bifactor model (which could be contrasted to, for example, a unidimensional model with method factors modelled through correlated residuals). I would consider an alternative label. Although the model was argued not be viable on the basis of the pattern of loadings with the method specific factors, I think readers would still be interested to know the loadings on the general factor. Additionally, for comparisons sake, I think readers could be interested in the fit of a pure unidimensional model (no specific factors).

The text refers sometimes to “Model 1” and “Model 2” etc. and I found it difficult to match these labels to the model structures. You could consider putting “Model 1: Bifactor gwb” etc in Table 1 to aid readers.

In relation to the bifactor models, it could be informative to calculate and report bifactor indices such as ωH and ECV. These would offer deeper insights into the uni- vs. multi-dimensionality of the general factors, and by extension issues such as the feasibility of calculating and interpreting total scores across items.

As a small detail, the specific factors for the bifactor gwb with method factor model are referred to as “gwb-free” to appropriately communicate the idea that specific factors explain variance after accounting for the general factor. In terms of interpretation, it is important to communicate that in this model the internalizing and externalizing specific factors are not only ‘gwb-free’, but also ‘method-free’ because they also load on the method factor.

After testing the competing measurement models, the authors used SEM to test a series of partial mediation models, each time using a different measurement model (correlated factors model, bifactor p model, and the preferred bifactor gwb model). The mediation model with the preferred bifactor gwb model had the best fit. When reporting the results from this model, the authors also present in the text the associations between the gwb-free specific factors, highlighting as a particular point of interest the positive association between the gwb-free Wellbeing specific factor and concurrent and future impairment. While these results must be presented (as they are in Table 3), I am uncertain whether it is necessary to elaborate them in the text given that it is often very difficult to interpret the meaning of specific factors after factoring out the general factor. As the authors rightly elaborate on in the discussion, “the gwb-free wellbeing specific factor is more difficult to interpret”, meaning ultimately it may well only represent a nuisance effect.

Discussion

“The shared underlying construct (gwb) between wellbeing and psychopathology appears to be a generalized negative mental state”. Firstly, as wellbeing items negatively loaded on the general factor I feel it is misleading to refer to the construct as ‘wellbeing’ in this sentence. I suggest using terminology that better captures this modelled relationship, such as (lack of) wellbeing. I note there are many instances where this is done already in the manuscript, such as in the discussion, but it must be done consistently.

Second, the discussion lacks consideration of what the general factor might represent other than a “generalized negative mental state”. It could be interesting to explore what this general factor may capture in terms of more stable higher-order, tendencies for illbeing and psychopathology. As an example, one could refer to Cloninger’s psychobiological model of personality for this purpose.

**Do you want your identity to be public for this peer review?** For information about this choice, including consent withdrawal, please see our Privacy Policy

Reviewer #1: No

---

## [Author Response · Author response to Decision Letter 1]

11 Oct 2025

RESPONSE TO REVIEWER COMMENTS

Reviewer #1

Thank you for the opportunity to review the article “Incorporating wellbeing into general factor models: a more complete mental state?”. The paper is thorough, well presented, and addresses an interesting topic within the literature, namely whether global mental health status might be modelled more accurately through the inclusion of wellbeing alongside psychopathology. The very large sample size, recruited across schools in six different regions of the UK, suggests sufficient statistical power to test the proposed measurement and structural models.

The authors would like to thank the reviewer for their positive feedback as well as their constructive comments for improvement.

The method for dealing with missing data was convincing. However, as a small detail, it was unclear to me from the text how the models were fit using the multiple imputed datasets. Was this using the lavaan.mi function? I think readers would benefit if this final part of the imputation process (e.g. pooling) was also explained.

Thank you to the reviewer for pointing this out. This has now been added to the manuscript (Line 305 - 310).

It would be of interest for readers to know the unidimensional internal consistency reliability (eg., ω or α) of the measures, with confidence intervals, in the study sample.

We conducted this as well as another additional analyses (fitting a unidimensional model) on an archived and fully anonymized version of the original dataset (http://doi.org/10.5255/UKDA-SN-9150-1). The original, identifiable, dataset has been destroyed due to ethical requirements. While the archived, fully anonymised version is largely similar to the original dataset, we would like to note that:

• The final sample size in the archived data is n = 15255 rather than n = 15258, reflecting minor data removal during anonymisation and preparation for public archiving.

• Several variables originally obtained through linkage with the National Pupil Database (ethnicity, free school meals (FSM) status, special educational needs (SEN) status, and IDACI (area deprivation) score) are not available in the archived dataset.

These differences do not affect the internal consistency reliability analysis, as this was conducted on the sample without multiple imputation, and the results are expected to be identical to those that would have been obtained with the original dataset. However, the other requested analyses below are conducted on multiply imputed data and the results are expected to vary very slightly from the original analyses (mostly, at the level of the 2nd or 3rd decimal point) because of the presence of 4 less socio-demographic variables during the imputation.

These additional analyses are reported in the Supplemental Material (S7: Additional analyses), with a detailed explanation.

The paper tested several competing measurement models that included indicators of psychopathology and wellbeing. A “bifactor-p” model without wellbeing items was tested as a comparison. The last model tested was labelled as “Unidimensional with method”, which I found misleading, because this is still a bifactor model (which could be contrasted to, for example, a unidimensional model with method factors modelled through correlated residuals). I would consider an alternative label.

We agree with the reviewer on this point. We have now updated the name to “bifactor with method factors” in the manuscript.

Although the model was argued not be viable on the basis of the pattern of loadings with the method specific factors, I think readers would still be interested to know the loadings on the general factor.

We have now updated the manuscript with the relevant details (Line 520 - 521). We also note that all the factor loadings (for both the general and method factors) are also available in S3 Table E.

Additionally, for comparisons sake, I think readers could be interested in the fit of a pure unidimensional model (no specific factors).

We specified the pure unidimensional model on the archived dataset, as reported above. The pure unidimensional did not show good fit, as has been the case in previous work (Patalay et al., 2015; Castellanos-Ryan et al., 2016).

We have reported this in the main manuscript (Line 530 - 538), specifying that it is on an archived dataset with further information about comparisons between original and archived datasets in S7: Additional analyses.

The text refers sometimes to “Model 1” and “Model 2” etc. and I found it difficult to match these labels to the model structures. You could consider putting “Model 1: Bifactor gwb” etc in Table 1 to aid readers.

We have now added the numbered labels to Table 1. We have also updated the text (e.g. Line 539) to ensure clarity while referring to models by their numbers.

In relation to the bifactor models, it could be informative to calculate and report bifactor indices such as ωH and ECV. These would offer deeper insights into the uni- vs. multi-dimensionality of the general factors, and by extension issues such as the feasibility of calculating and interpreting total scores across items.

We have calculated omega hierarchical and explained common variance for the bifactor models fit on the original dataset. For omega hierarchical, we used a custom function that substantively replicated compRelSEM’s (semTools) workflow to calculate this. Since our models included negative loadings of SWEMWEBS items on the general factor, directly using compRelSEM gave incorrect results for the reliability of the general factor, hence we had to use a custom function to replicate compRelSEM. For explained common variance, we used a custom function based on the BifactorIndices package, applied to pooled standardized factor loadings. Both pipelines used are available on OSF.

As a small detail, the specific factors for the bifactor gwb with method factor model are referred to as “gwb-free” to appropriately communicate the idea that specific factors explain variance after accounting for the general factor. In terms of interpretation, it is important to communicate that in this model the internalizing and externalizing specific factors are not only ‘gwb-free’, but also ‘method-free’ because they also load on the method factor.

We thank the reviewer for pointing this out. We have updated the manuscript to refer to these specific factors as gwb-free and method-free.

After testing the competing measurement models, the authors used SEM to test a series of partial mediation models, each time using a different measurement model (correlated factors model, bifactor p model, and the preferred bifactor gwb model). The mediation model with the preferred bifactor gwb model had the best fit. When reporting the results from this model, the authors also present in the text the associations between the gwb-free specific factors, highlighting as a particular point of interest the positive association between the gwb-free Wellbeing specific factor and concurrent and future impairment. While these results must be presented (as they are in Table 3), I am uncertain whether it is necessary to elaborate them in the text given that it is often very difficult to interpret the meaning of specific factors after factoring out the general factor. As the authors rightly elaborate on in the discussion, “the gwb-free wellbeing specific factor is more difficult to interpret”, meaning ultimately it may well only represent a nuisance effect.

We had initially described these in the text for transparency rather than substantial interpretation. We have now reduced this elaboration both in the Results and Discussion section, especially for the wellbeing factor, while retaining some discussion about the Int and Ext factors, as they seem relatively more reliable.

The shared underlying construct (gwb) between wellbeing and psychopathology appears to be a generalized negative mental state”. Firstly, as wellbeing items negatively loaded on the general factor I feel it is misleading to refer to the construct as ‘wellbeing’ in this sentence. I suggest using terminology that better captures this modelled relationship, such as (lack of) wellbeing. I note there are many instances where this is done already in the manuscript, such as in the discussion, but it must be done consistently.

We have now carefully read through the manuscript and edited this for consistency throughout.

Second, the discussion lacks consideration of what the general factor might represent other than a “generalized negative mental state”. It could be interesting to explore what this general factor may capture in terms of more stable higher-order, tendencies for illbeing and psychopathology. As an example, one could refer to Cloninger’s psychobiological model of personality for this purpose.

We have been deliberately tempered in our substantive interpretations of the general factor, in light of recent critiques of etiological interpretations of general factor modelling. However, in agreement with the reviewer’s point, we have now balanced this and added a section to the manuscript (Line 700-712) reflecting on what our gwb factor might represent.

Editor’s comments

This study has a large sample size, employs innovative methods, and presents a well-structured introduction. The data provided in the supplementary materials are also robust.

However, I recommend standardizing the format of the p-values to avoid using scientific notation.

We have updated this in the supplementary materials to reflect p-value thresholds (<0.05, <0.01 and <0.001) instead of scientific notation.

In addition, I suggest expanding the discussion section to include more practical implications of the methods used, which would help other researchers better understand and apply them.

We have now added implications of using structural equation modelling to the manuscript in Lines 763 – 774, especially how they can be differentiated from sum score approaches. Further, Line 862-872 also contains practical implications of using structural equation modelling when measuring mental health status.

Additional Requirements

We have gone through the manuscript and corrected all stylistic issues we could find. Please let us know if there is anything we have missed.

We would like to clarify that an anonymized version of the dataset used in this study is available publicly at: http://doi.org/10.5255/UKDA-SN-9150-1. However, due to the process of anonymisation as well as the ethical restrictions on data storage, the archived dataset does not include four variables (IDACI score, ethnicity, Free School Meals (FSM) status, and Special Educational Needs (SEN) status) and contains three fewer participants than the original study sample used for the analyses in this paper. These four variables were not part of the original survey; they were derived through linkage with the National Pupil Database (NPD). Under current security standards, NPD-linked data can only be archived in specific secure platforms (such as the ONS Secure Research Service) and therefore could not be deposited in the UK Data Service. Therefore, the survey results without the NPD-linked variables were archived in the UK Data Service. Nevertheless, we believe that the substantive conclusions of this study should be able to be replicated by interested readers using this dataset and our methods, as we have demonstrated in our revised manuscript as well.

We have added two references which were missing from the reference list.

References

1. Castellanos-Ryan, Natalie, Frederic N. Brière, Maeve O’Leary-Barrett, et al. ‘The Structure of Psychopathology in Adolescence and Its Common Personality and Cognitive Correlates.’ Journal of Abnormal Psychology 125, no. 8 (2016): 1039–52. https://doi.org/10.1037/abn0000193.

2. Patalay, Praveetha, Peter Fonagy, Jessica Deighton, Jay Belsky, Panos Vostanis, and Miranda Wolpert. ‘A General Psychopathology Factor in Early Adolescence’. British Journal of Psychiatry 207, no. 1 (2015): 15–22. https://doi.org/10.1192/bjp.bp.114.149591.

---

## [Editor Report · Decision Letter 1]

14 Oct 2025

Incorporating wellbeing into general factor models: a more complete mental state?

PONE-D-25-20026R1

Dear Dr. Ritika Chokhani,

We’re pleased to inform you that your manuscript has been judged scientifically suitable for publication and will be formally accepted for publication once it meets all outstanding technical requirements.

Kind regards,

Zheng Zhang

Academic Editor

PLOS ONE

Additional Editor Comments (optional):

The quality of the manuscript has improved after revision; acceptance is recommended.
---

## [Editor Report · Acceptance letter]

PONE-D-25-20026R1

PLOS ONE

Dear Dr. Chokhani,

I'm pleased to inform you that your manuscript has been deemed suitable for publication in PLOS ONE. Congratulations! Your manuscript is now being handed over to our production team.

Kind regards,

on behalf of

Dr. Zheng Zhang

Academic Editor

PLOS ONE